# Effects of Recreational Small-Sided Soccer Games on Bone Mineral Density in Untrained Adults: A Systematic Review and Meta-Analysis

**DOI:** 10.3390/healthcare9040457

**Published:** 2021-04-13

**Authors:** Filipe Manuel Clemente, Rodrigo Ramirez-Campillo, Hugo Sarmento, Daniel Castillo, Javier Raya-González, Thomas Rosemann, Beat Knechtle

**Affiliations:** 1Escola Superior Desporto e Lazer, Instituto Politécnico de Viana do Castelo, Rua Escola Industrial e Comercial de Nun’Álvares, 4900-347 Viana do Castelo, Portugal; 2Instituto de Telecomunicações, Delegação da Covilhã, 1049-001 Lisboa, Portugal; 3Human Performance Laboratory, Department of Physical Activity Sciences, Universidad de Los Lagos, Lord Cochrane 1046, Osorno 5290000, Chile; r.ramirez@ulagos.cl; 4Centro de Investigación en Fisiología del Ejercicio, Facultad de Ciencias, Universidad Mayor, Santiago 7500000, Chile; 5University of Coimbra, Coimbra 3004-531, Portugal; hg.sarmento@gmail.com; 6Research Unit for Sport and Physical Activity, Faculty of Sport Sciences and Physical Education, 3004-531 Coimbra, Portugal; 7Faculty of Health Sciences, Universidad Isabel I, 09003 Burgos, Spain; danicasti5@gmail.com (D.C.); rayagonzalezjavier@gmail.com (J.R.-G.); 8Institute of Primary Care, University of Zurich, 8091 Zurich, Switzerland; thomas.rosemann@usz.ch; 9Medbase St. Gallen Am Vadianplatz, 9001 St. Gallen, Switzerland

**Keywords:** sports, football, bone mass, recreational football, health promotion, human physical conditioning

## Abstract

This systematic review with meta-analysis was conducted to assess the effects of small-sided games (SSG)-based training programs on bone mineral density (BMD) in untrained adults. The data sources utilized were Cochrane, Embase, Medline (PubMed), Scopus, SPORTDiscus, and Web of Science. The study eligibility criteria were: (i) untrained adults (>18 years old) of any sex, with or without a noncommunicable disease; (ii) SSG-based programs with a minimum duration of four weeks and no restrictions regarding frequency (number of sessions per week); (iii) passive or active control groups; (iv) pre-post values of BMD; (v) only randomized controlled trials; and (vi) only original and full-text studies written in English. The database search initially yielded 374 titles. From those, nine articles were eligible for the systematic review and meta-analysis. The age of included population varied from a minimum of 20 and a maximum of 71 years old. Non-significant differences between SSG and passive and active control groups on total BMD (ES = 0.14; *p =* 0.405 and ES = 0.28; *p =* 0.05, respectively). Meanwhile, significant differences in favor of SSGs vs. passive and control groups were detected, evidencing an improvement of BMD in lower limbs of the adult population for both sexes (ES = 0.26; *p =* 0.05 and ES = 0.28; *p =* 0.156, respectively). As conclusions, SSGs can be used as a non-pharmacological alternative to increase the BMD in the lower limbs despite having no significant impact on total body BMD. Careful generalization should be done of the level of heterogeneity.

## 1. Introduction

Bone remodeling in adulthood involves the processes of bone resorption and formation, which occur in a continuous process in which bone tissue is removed and replaced by bone cells [1]. An imbalance in this process may cause the loss of bone mass and, ultimately, osteopenia or osteoporosis [2].

Mechanotransduction is one of the main mechanisms that contribute to bone remodeling. This mechanism consists of a conversion of a mechanical force into a cellular response, as proposed by Wolff’s law [3,4]. Osteocytes within the bone and lining cells on the bone surface are the first mechanosensors in bone tissue to detect mechanical strains and deformation on bone matrix [5]. After such deformation, the osteocytes send paracrine signals to osteoblasts and osteoclasts [5]. Thus, mechanical load and strain promoted in the body will stimulate the participation of osteoblasts and osteoclasts (bone cells) in bone remodeling (i.e., formation and resorption) [6]. Disuse of lack of loading causes an acceleration of bone turnover with bone resorption overwhelming bone formation and conducting to a faster loss of bone mass [5]. This is one of the reasons why exercise is an effective non-pharmacological approach to increase the health of bones, considering the mechanical stimulus provided to the bone [7,8]. Naturally, the specific effects on bone structure will depend on the frequency, duration, magnitude, and rate of loading promoted [9].

Different approaches can be used to analyze bone health, although bone mineral density (BMD) is often used as the sole measure representing bone strength [10]. It is also one of the most important predictors of fracture risk in both men and women [11,12]. BMD represents the amount of bone mineral in bone tissue, thus providing an idea of the mass of mineral per volume of bone (density). Usually, this measure is determined by densitometry [13].

Research on the effects of exercise on adult BMD is consistent among some specific populations, such as postmenopausal women [14,15], premenopausal women [16], and older men [8]. Although some meta-analyses suggest that exercise’s beneficial effects on BMD are independent of the type of exercise in postmenopausal women [14], it seems that specific exercises (e.g., swimming or cycling) are not meaningfully effective in menopausal women [17] or even in children and adolescents [18]. This can be justified by the fact that high mechanical impact should be implemented to signalize bone remodeling and stimulate the mineralization and synthesis of the bone matrix [19]. Additionally, it also seems that the effectiveness of different types of exercise can vary in different regions of the body (e.g., BMD of hip, spine, or femur) [16].

Considering that exercise can be an effective strategy for improving bone health, it is important to find strategies to maximize the practice’s adherence [20]. Nowadays, sedentary behaviors and lack of physical activity and exercise is a major issue [21] that can be mitigated by finding exercise alternatives that improve pleasure and enjoyment [22]. In the particular case of adults, some strategies for increasing adherence to exercise have been proposed—including using popular games, such as soccer—to increase the enjoyment of untrained or sedentary populations to regular practice [23]. Within recreational soccer, small-sided games (SSGs) are very popular drills since participants perform more actions than they would in a match. The SSGs are drill-based exercises, in which the dynamics of the formal game are simplified by reducing the format of play (e.g., 2vs.2, 3vs.3), changing the pitch configuration (e.g., pitch dimensions, length: width ratio), or changing other task constraints related to actions, objectives or behaviors [24]. These task adjustments are implemented to differentiate physiological and physical stimuli [25,26].

The use of recreational soccer as an approach for non-pharmacological clinical programs for specific adult populations has been widely researched [27,28,29]. However, the use of SSGs, in particular, was only recently summarized in a systematic review that was not dedicated to a specific outcome, and no meta-analysis was conducted [30]. Therefore, there remains a need for a meta-analytical comparison that provides information about the effectiveness of recreational soccer SSGs vs. control groups on BMD in adult populations. This will improve our understanding of the potential use of these programs to benefit adults—specifically those who need a non-pharmacological approach to improve their bone health.

Due to the absence of a systematic review and meta-analysis of the effects of soccer SSGs on the BMD of untrained adults, this study aimed to assess the effects of SSG-based programs on the BMD of untrained men and women.

## 2. Materials and Methods

This systematic review and meta-analysis followed the Cochrane Collaboration guidelines [31]. The systematic review strategy was conducted according to preferred reporting items for systematic reviews and meta-analyses (PRISMA) guidelines [32]. The PICOS approach (population, intervention, comparator, outcomes, study design) was followed: (P) untrained adults (>18 years old) from any sex, with or without a noncommunicable disease; (I) SSG-based programs with a minimum of 4 weeks of intervention and no restricted to frequency; (C) passive or control groups no exposed to a specific pharmacological or diet-oriented plan; (O) bone mineral density measured in any body part; and (S) randomized controlled trials. The protocol was registered with the International Platform of Registered Systematic Review and Meta-Analysis Protocols with the number 202,110,095 and the DOI number 10.37766/inplasy2021.10095.

### 2.1. Eligibility Criteria

Inclusion and exclusion criteria for this systematic review and meta-analysis can be found in Table 1.

A reference manager software was used to identify the duplicates. A screening process for the title, abstract and reference list was made for each study to locate potentially relevant studies (made by authors HS and JRG). Both authors also reviewed the full version of the papers in detail to identify articles that met the selection criteria and those that were excluded. A third author (DC) participated in a discussion for eventual discrepancies regarding the selection process.

### 2.2. Information Sources

Electronic databases (Cochrane, Embase, PubMed, Scopus, SPORTDiscus, and Web of Science) were searched for relevant publications from inception up to 5 April 2021. Keywords and synonyms were entered in various combinations in all fields: (“soccer” OR “football”) AND (“soccer training” OR “football training” OR “soccer game*” OR “conditioned game*” OR “small-sided soccer game*” OR “small-sided and conditioned game*” OR “SSG”) AND (“bone mineral density” OR “bone mass” OR “BMD”). Additionally, the reference lists of the included studies retrieved were manually searched to identify potentially eligible studies not captured by the electronic searches. Finally, an external expert was contacted to verify the final list of references included in this systematic review and indicate any study that was not detected through our search.

### 2.3. Extraction of Data

An Excel data sheet (Microsoft Corporation, Redmond, WA, USA) was prepared following Cochrane Consumers and Communication Review Group [33] to assess inclusion requirements and subsequently tested on ten randomly selected studies (i.e., pilot testing). The assessing process was performed by two independent authors (JRG and DC). Disagreements about the study eligibility were discussed with a third author (FMC) to achieve a consensus. Full-text articles excluded, with reasons, were recorded.

### 2.4. Data Items

To establish consistency in data analyzing and reporting, only measures that were analyzed three or more times for different articles were included. The BMD (g·cm^−2^) was chosen as the main outcome for the following body regions: total body (or whole-body), spine (or trunk or midriff), pelvis (or hips) and lower limb (leg, femur and tibia). These regions were extracted based on the criteria, including measures analyzed three times or more in different articles. The pre- and post-data were extracted. Intermediate assessments (in the middle of interventions) and/or follow-up periods (without intervention) were not extracted. The method for assessing the BMD was also extracted. Adverse effects were also extracted in case of any reported. Additionally, the following information was extracted from the included studies: (i) number of participants (n), age (years), sex and clinical condition (noncommunicable disease); (ii) the SSGs format and pitch size (if available); (iii) period of intervention (maximum number of weeks in intervention) and number of sessions per week (n/w); and (iv) regimen of intervention (work duration, work intensity, modality, relief duration, relief intensity, repetitions and series, between-set recovery).

### 2.5. Assessment of Methodological Quality

The physiotherapy evidence database (PEDro) scale was used to assess the methodological quality of the randomized controlled trials included in this systematic review and meta-analysis. The PEDro scale scores the internal study validity in a range of 0 (high risk of bias) to 10 (low risk of bias). Eleven items are measured on the scale. Criterion 1 is not included in the final score. Points for items 2 to 11 were only attributed when a criterion was clearly satisfied. Two of the authors (JRG and DC) independently scored the articles. Disagreements in the rating between both authors were resolved through discussion with a third author (HS). To control the risk of bias between authors, a kappa correlation test was used to analyze the agreement level for the included studies. The agreement level of k = 0.86 was obtained.

### 2.6. Summary Measures, Synthesis of Results, and Publication Bias

Although two studies can be used in meta-analyses [34], considering reduced sample sizes are common in the sports science literature [35], including in SSG studies [30], analysis and interpretation of results in this systematic review and meta-analysis were only conducted in the case of at least three studies provided baseline and follow-up data for the same measure. Pre-training and post-training means and standard deviations (SD) for dependent variables were used to calculate effect sizes (ES; Hedge’s *g*) for each outcome measure in the SSG-based training and control groups. Data were standardized using post-intervention SD values. The random-effects model was used to account for differences between studies that may impact the SSG-based effect [36,37]. The ES values are presented with 95% confidence intervals (CI). Calculated ES were interpreted using the following scale: <0.2, trivial; 0.2–0.6, small; >0.6–1.2, moderate; >1.2–2.0, large; >2.0–4.0, very large; >4.0, extremely large [38]. Heterogeneity was assessed using the *I*^2^ statistic, with values of <25%, 25–75%, and >75% considered to represent low, moderate, and high heterogeneity levels, respectively [39]. The risk of bias was explored using the extended Egger’s test [40]. When bias was present, the trim and fill method was applied [41], in which case L0 was assumed as the default estimator for missing studies [42]. All analyses were carried out using the Comprehensive Meta-Analysis software (version 2; Biostat, Englewood, NJ, USA). Statistical significance was set at *p* ≤ 0.05.

## 3. Results

### 3.1. Study Identification and Selection

The searching of databases identified an initial 374 titles. These studies were then exported to reference manager software (EndNote^TM^ X9, Clarivate Analytics, Philadelphia, PA, USA). Duplicates (90 references) were subsequently removed either automatically or manually. The remaining 284 articles were screened for their relevance based on titles and abstracts, resulting in removing a further 236 studies. The full texts of the remaining 48 articles were examined diligently. After reading full texts, a further 39 studies were excluded owing to a number of reasons, including youth population (n = 19); conference abstracts (n = 8); other languages than English (n = 4), including data from other sports (n = 4); lack of control group (n = 2); repeated data (n = 1); lack of pre-post mean and standard deviation data (n = 1). Therefore, 9 articles that provided mean and standard deviation post-training data for at least the main outcome were eligible for the systematic review and meta-analysis (Figure 1).

### 3.2. Study Characteristics

The characteristics of the ten studies included in the meta-analysis can be found in Table 2. Additionally, the details of the SSG-based programs can be found in Table 3. Finally, characteristics of the control groups can be found in Table 4. The included randomized-controlled studies involved 9 individual experimental groups and 163 participants, 7 active control groups with 118 participants and 9 passive control groups with a total of 140 participants.

### 3.3. Methodological Quality

The quality scores of each study using the PEDro checklist are displayed in Table 5. The mean score was 5.78 (minimum: 5, maximum: 7). None of the studies reported that subjects, therapists or consultants were blinded, and only one study indicated that allocation was concealed. All studies conducted an intent-to-treat analysis, between-group analyses and provided point estimates for effect size, while three studies failed in terms of >85% of participants completed the intervention.

### 3.4. SSG vs. Control: Effects on Total Body BMD

A summary of the included studies and total body BMD results reported before and after SSG-based intervention and control groups are provided in Table 6.

Eight studies provided data for total body BMD, involving eight experimental and eight passive control groups (pooled *n* = 282). Results showed a trivial effect of SSGs on total body BMD (ES = 0.14; 95% CI = −0.19 to 0.47; *p =* 0.405; *I^2^* = 46.3%; Egger’s test *p* = 0.238; Figure 2) when compared to passive controls. For the assessment of publication bias, a funnel plot is presented in Figure 3.

Five studies provided data for total body BMD, involving five experimental and six active control groups (pooled *n* = 181). Results showed a small effect of SSGs on total body BMD (ES = 0.28; 95% CI = −0.11 to 0.68; *p =* 0.156; *I^2^* = 39.0%; Egger’s test *p* = 0.100; Figure 4) when compared to active controls. For the assessment of publication bias, a funnel plot is presented in Figure 5.

Eight studies provided data for total body BMD, involving eight experimental groups (pooled *n* = 151). Results showed a small effect of SSGs on total body BMD (ES = 0.23; 95% CI = −0.01 to 0.46; *p =* 0.057; *I*^2^ = 70.2%; Egger’s test *p* = 0.009; Figure 6). After the Trim and Fill method was applied, the adjusted values remained equal to the observed values. For the assessment of publication bias, a funnel plot is presented in Figure 7.

### 3.5. SSG vs. Control: Effects on Spine BMD

A summary of the included studies and results of spine BMD reported before and after SSG-based intervention and control groups are provided in Table 7. Among the included studies, after–before percent variations in SSG varied from −0.9 to −4.3%, while passive controls ranged from −2.2 and −0.9%. In the active control (Zumba), there was no change between assessments.

### 3.6. SSG vs. Control: Effects on Pelvis BMD

A summary of the included studies and pelvis BMD results reported before and after SSG-based intervention and control groups are provided in Table 8. Among the SSG groups, changes ranged −0.9 and −1.9%, while active control ranged −1.0 and −3.6% and in passive control, between −0.9 and −1.8%.

### 3.7. SSG vs. Control: Effects on Lower Limb BMD

A summary of the included studies and results of lower limb BMD reported before and after SSG-based intervention and control groups are provided in Table 9.

Seven studies provided data for lower body BMD, involving seven experimental and seven passive control groups (pooled *n* = 226). Results showed a small effect of SSGs on lower body BMD (ES = 0.26; 95% CI = 0.00 to 0.51; *p =* 0.05; *I*^2^ = 0.0%; Egger’s test *p* = 0.989; Figure 8) when compared to passive controls. For the assessment of publication bias, a funnel plot is presented in Figure 9.

Five studies provided data for lower body BMD, involving five experimental and six active control groups (pooled *n* = 195). Results showed a small effect of SSGs on lower body BMD (ES = 0.28; 95% CI = 0.00 to 0.56; *p =* 0.05; *I^2^* = 0.0%; Egger’s test *p* = 0.785; Figure 10) when compared to active controls. For the assessment of publication bias, a funnel plot is presented in Figure 11.

Seven studies provided data for lower body BMD, involving seven experimental groups (pooled *n* = 119). Results showed a small effect of SSGs on lower body BMD (ES = 0.42; 95% CI = 0.12 to 0.72; *p =* 0.006; *I^2^* = 74.6%; Egger’s test *p* = 0.035; Figure 12). After the Trim and Fill method was applied, the adjusted values was ES = 0.34 (95% CI = 0.04 to 0.64). For the assessment of publication bias, a funnel plot is presented in Figure 13.

## 4. Discussion

The main aim of this systematic review with meta-analysis was to assess the effects of soccer SSG programs on BMD in untrained adults. Although non-significant differences between SSG and control groups were found in terms of total body BMD, significant differences in favor of SSGs were detected in the lower limbs of adults of both sexes. All included studies were categorized as being of fair or high methodological quality, which, therefore, strengthens these conclusions.

### 4.1. Effects of SSGs-Based Intervention on Total Body BMD

The importance of BMD across all ages and sexes is widely recognized (Johnell et al. 2005; Turner 2002) since this variable is a key predictor of future potential diseases [52]. As such, one of the main goals of physical and conditioning specialists of high-risk populations (e.g., untrained adults) is to maintain or improve BMD levels through entertaining activities that encourage contact with peers, as recreational soccer does [28]. Our meta-analysis results revealed non-significant differences in the increase of total body BMD when the effects of soccer SSG programs were compared with passive control conditions. In this regard, although recreational soccer seems to be an interesting strategy due to its psychosocial benefits [53], it could not provide enough stimulation to improve total body BMD in untrained adults. Our results do not support those reported by Krustrup et al. [46], which observed improvements in total body BMD after 16 months of SSG intervention. Nevertheless, most studies on this topic did not report significant differences [43,44,47,49]. This could be influenced by the duration of training interventions (from 12 weeks to 16 months), although further studies are necessary. On the other hand, the sample’s heterogeneity included in this meta-analysis could have affected the results since studies involving only men, only women, or both and using specific samples in terms of typology (with or without a noncommunicable disease) were included.

Likewise, no significant differences in BMD were observed when comparing the SSG program group and active groups. Our results support the findings observed in all the included studies. In this sense, and despite the different activities that were included in the active control condition (e.g., resistance training, Zumba, swimming) [43,48], none of the included studies reported significant differences between these activities and soccer SSG programs, showing that these sport modalities could have similar effects on BMD. Thus, the preferences and necessities of each population must be considered when developing a physical activity program designed to maintain BMD. Additionally, when subjects need to increase their BMD due to pathological conditions, other training configurations must be proposed [54].

Finally, no significant within-group changes in total body BMD were reported in our meta-analysis, although a *p*-value near to significance was obtained (*p* = 0.057). This could be because most of the included investigations [44,47,49,50] reported non-significant gains in total body BMD, while only one showed a significant effect of soccer SSGs on this BMD variable [46]. The remaining studies [43,48,51] reported that total body BMD levels were maintained after the soccer interventions, a finding that could be useful for healthy subjects that do not need to increase their BMD levels but simply maintain it to avoid future diseases.

Considering the aforementioned findings, recreational soccer seems to have a positive osteogenic impact, which could be considered as an interesting strategy to maintain total body BMD in untrained adults and reduce the negative age symptoms related to bone health, which could lead to falls, bone fractures, and diseases [47]. Further, it would be advisable to engage in long-term (i.e., at least 12 months) recreational soccer interventions to increase the total body BMD of untrained adults.

### 4.2. Effects of SSGs-Based Intervention on Lower Limb BMD

Soccer is a multidirectional sport characterized by the presence of many-body impacts of different magnitudes [55], which mainly involve the participants’ lower limbs [56]. While our meta-analysis revealed no improvements in total body BMD, greater improvements in lower limb BMD were obtained for SSG groups compared to passive control groups. Our results are in line with several previous studies [43,44,45,48,49], which observed improvements in lower limb BMD after different time-period interventions (ranging from 12 weeks to 64 months). These changes could be influenced by the specificity of soccer interventions concerning the BMD variable assessed (i.e., lower limb BMD). Specifically, these improvements in favor of SSG groups could be supported by the muscles involved during soccer practice (i.e., the hamstring, adductor, abductor, and quadriceps). Such improvements may also be influenced by some physiological mechanisms—such as a pronounced increase in plasma bone turnover markers, an elevation of the plasma osteocalcin, or a significant increase in the procollagen type-1 amino-terminal propeptide [44]. Therefore, the impacts of running and intensity actions occurring in SSGs may conduct enough mechanical load and strain stimulus, thus promoting the participation of osteoblasts and osteoclasts (bone cells) in bone remodeling (i.e., formation and resorption) [6] as a consequence of mechanotransduction. Considering that tensions generated by muscular contraction may stimulate bone cells with strain stress, compression force, and shear stress [57], it is possible that changes of direction, accelerations/decelerations, passing or kicking occur in SSGs often, can generate a localized positive stimulus for bone formation.

To further explore lower limb BMD, significant differences were also observed when compared soccer SSG programs with active control conditions. These results coincide with those previously observed in the literature [43,45,46], where were implemented activities in which impacts were encountered (i.e., Zumba and running) for control groups. In those studies that compared SSG groups with active controls, participants who experienced soccer practice obtained better benefits in lower limb BMD, mainly due to the characteristics of the soccer-based impacts. This means that high mechanical impact is generated by large ground reactions and muscle forces exerted on the bone tissue at a high strain rate and in varied directions in relation to the longitudinal bone axis), which seems to align with the definition of an optimal osteogenic stimulus [58]. This explanation becomes more relevant when comparing SSGs with no-impact activities (i.e., swimming), which, even if performed for a long time, could result in reduced lower limb BMD [48]. Despite the superior effects of SSGs in increasing lower limb BMD levels in untrained adults of both sex, further research involving different control activities still needs to be implemented to prescribe appropriate training exercises.

Following the line of aforementioned findings, significant within-group variations in lower limb BMD were reported in our meta-analysis, mainly because five of seven studies analyzed in this category reported significant increases in lower limb BMD after SSG-based interventions [43,44,45,48,49]. Specifically, it was observed that younger populations (i.e., 30–45 years old) exhibited a higher osteogenic response to exercise than elderly individuals, showing that the inclusion of soccer SSGs programs could be a useful strategy for elderly participants. However, longer interventions would be advisable for achieving such benefits in lower limb BMD [44,45]. Thus, soccer SSGs are an interesting strategy for healthy individuals to prevent osteoporosis and bone fragility later in life [45], as well as to increase BMD through exercise regimens especially designed to maximize the osteogenic impact in elderly people (Krustrup et al. 2010).

### 4.3. Effects of SSGs-Based Intervention on Spine and Pelvis

In addition to total body and lower limb BMD, spine and pelvis BMD are considered relevant markers of bone diseases [59], mainly in some special populations (e.g., postmenopausal females). Nevertheless, scarce literature has been published based on soccer training in which the effects of SSGs on these markers have been analyzed. Specifically, two studies reported the effects of soccer SSGs on spine BMD, with both presenting non-significant improvements [43,50]. Skoradal et al. (2018) observed an improvement in spine BMD after 16 weeks of soccer training in pre-diabetic women, although non-significant differences with the control group (passive) were reported. On the other hand, Barene et al. (2014) did not find any significant differences when comparing SSGs’ effects on spine BMD among two control groups (passive and active). Additionally, these authors found that participants in the active control group (who performed Zumba activity) reported greater spine BMD improvements, suggesting a need to perform activities other than soccer to improve BMD in this body region [43].

Regarding pelvis BMD, two studies have analyzed the effects of soccer SSGs on this marker [48,50]. No significant differences were reported after 15- to 16-week interventions based on SSGs when compared to passive and active conditions, even though a significant decrease in pelvis BMD in the active control group was reported [48]. Additionally, Skoradal et al. (2018) performed an individualized analysis and reported better adaptations in unhealthy participants regarding pelvis BMD. Considering the scarce literature analyzing the effects of soccer SSGs on spine and pelvis BMD, it would be worthwhile to carry out more investigations that could prescribe different durations and protocols of intervention and active-comparator groups.

### 4.4. Study Limitations, Future Research and Clinical Implications

This systematic review with meta-analysis is not exempt from limitations. First, the heterogeneity sample included in the selected studies is noted by the range of age (from 30 to 71 years, sex (male or female), and the typology of the participants, as different populations presented different pathologies (e.g., prostate cancer or prediabetes) and included healthy adults. Second, only one long-term study (i.e., experimental period equal to or greater than more months) was included when total body BMD was meta-analyzed, so robust conclusions about the influence of the experimental extension are limited. Third, lower limb BMD was assessed using different markers (e.g., femur, tibia, or whole lower limb). Fourthly, the small number of studies related to the influence of soccer SSG programs on spine and pelvis BMD (i.e., only two investigations for each marker) suggest that strength and conditioning specialists should be cautious when drawing conclusions and implementing practical applications. Fifthly, the small number of studies included does not allow a consistent conclusion for generalization. Finally, to the limited number of studies, it was impossible to run subgroup analysis based on determinant moderators as sex, age, or exercise intensity.

Due to the promising benefits of SSG on lower limb BMD, future studies analyzing differences in this marker or other lower limb markers following different SSG formats would be interesting. Additionally, longer SSG programs need to be studied to validate our findings of the effects of soccer SSG programs on total body BMD.

Overall, the findings derived from our systematic review with meta-analysis suggest that soccer SSGs are a useful strategy for improving lower limb BMD in untrained adults, as they present better results than passive and active control groups. However, it seems pertinent to analyze the specific characteristics of each population to design effective soccer SSG programs. On the other hand, recreational soccer could be considered as part of an interesting strategy to maintain total body BMD in untrained adults due to its positive osteogenic impact.

## 5. Conclusions

The present systematic review and meta-analysis revealed a significant small beneficial effect of using SSGs when compared to control conditions (passive and active) for improving BMD in the lower limbs of adults of both sexes. However, non-significant differences between SSG and controls were detected in the effects in total body BMD. Possibly, a specificity of practice may justify these findings. Nevertheless, the small number of included studies and the heterogeneity should be emphasized, namely, to interpret the conclusion with caution. For this reason, more research is needed, specifically considering the different types of clinical populations analyzed in the included articles.

## Figures and Tables

**Figure 1 healthcare-09-00457-f001:**
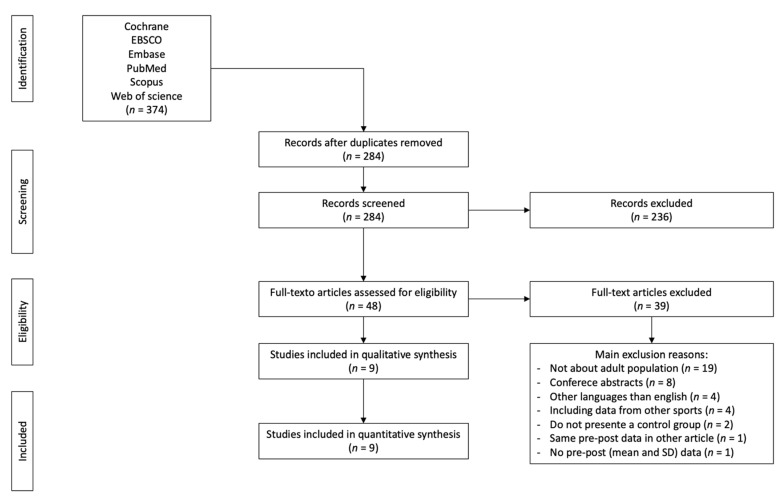
Preferred reporting items for systematic reviews and meta-analyses (PRISMA) flow diagram highlighting the selection process for the studies included in the current systematic review.

**Figure 2 healthcare-09-00457-f002:**
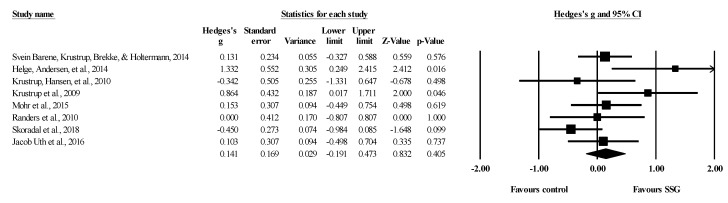
Forest plot of changes in total body bone mineral density in untrained adults after a training program based on soccer small-sided games (SSG) compared to a passive control condition. Values shown are effect sizes (Hedges’s g) with 95% confidence intervals (CI). The size of the plotted squares reflects the statistical weight of each study. The black diamond reflects the overall result.

**Figure 3 healthcare-09-00457-f003:**
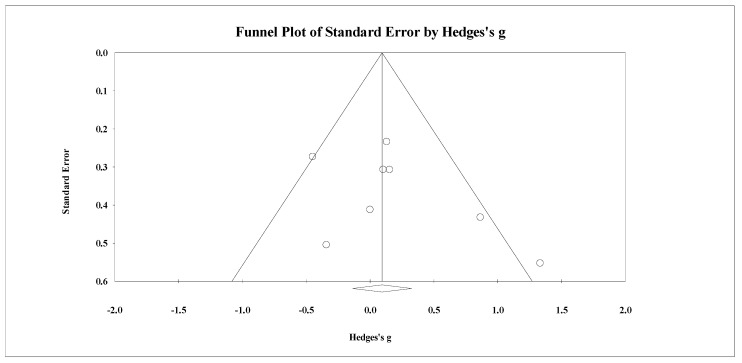
Funnel plot for changes in total body bone mineral density in untrained adults after a training program based on soccer small-sided games compared to a passive control condition. White circles: observed studies.

**Figure 4 healthcare-09-00457-f004:**
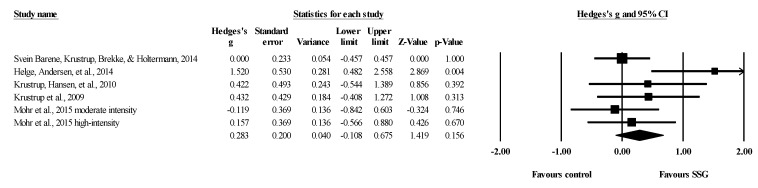
Forest plot of changes in total body bone mineral density in untrained adults after a training program based on soccer small-sided games (SSG) compared to active controls. Values shown are effect sizes (Hedges’s g) with 95% confidence intervals (CI). The size of the plotted squares reflects the statistical weight of each study. The black diamond reflects the overall result.

**Figure 5 healthcare-09-00457-f005:**
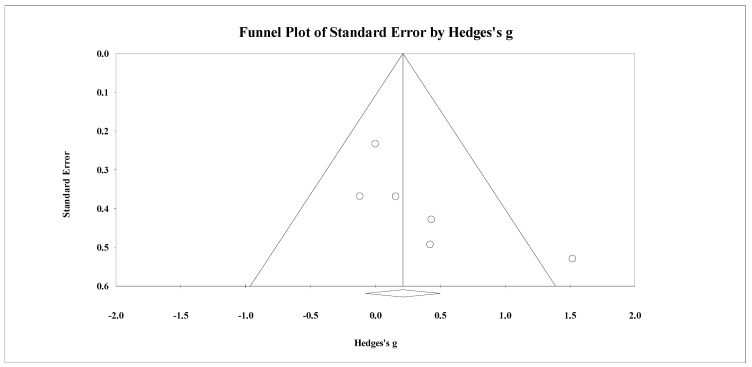
Funnel plot for changes in total body bone mineral density in untrained adults after a training program based on soccer small-sided games compared to an active control condition. White circles: observed studies.

**Figure 6 healthcare-09-00457-f006:**
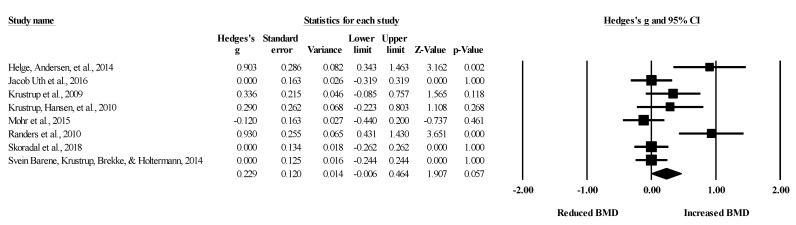
Forest plot of within-group changes in total body bone mineral density (BMD) in untrained adults after a training program based on soccer small-sided games. Values shown are effect sizes (Hedges’s g) with 95% confidence intervals (CI). The size of the plotted squares reflects the statistical weight of each study. The black diamond reflects the overall result.

**Figure 7 healthcare-09-00457-f007:**
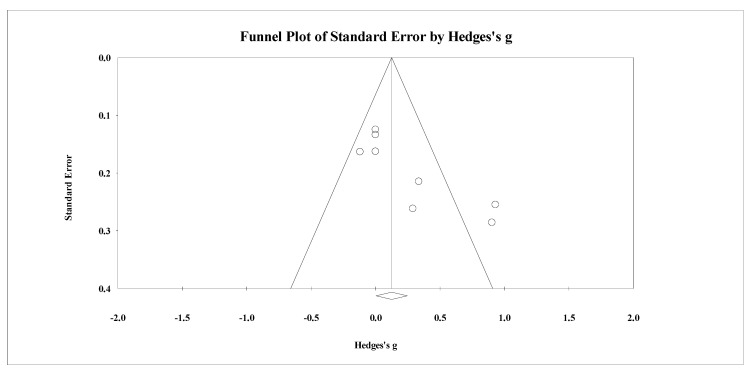
Funnel plot for changes in total body bone mineral density in untrained adults after a training program based on soccer small-sided games. White circles: observed studies.

**Figure 8 healthcare-09-00457-f008:**
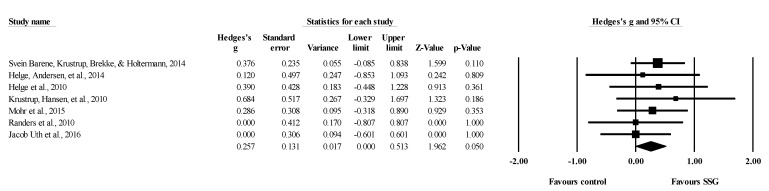
Forest plot of changes in lower-body bone mineral density in untrained adults after a training program based on soccer small-sided games (SSG) compared to controls. Values shown are effect sizes (Hedges’s g) with 95% confidence intervals (CI). The size of the plotted squares reflects the statistical weight of each study. The black diamond reflects the overall result.

**Figure 9 healthcare-09-00457-f009:**
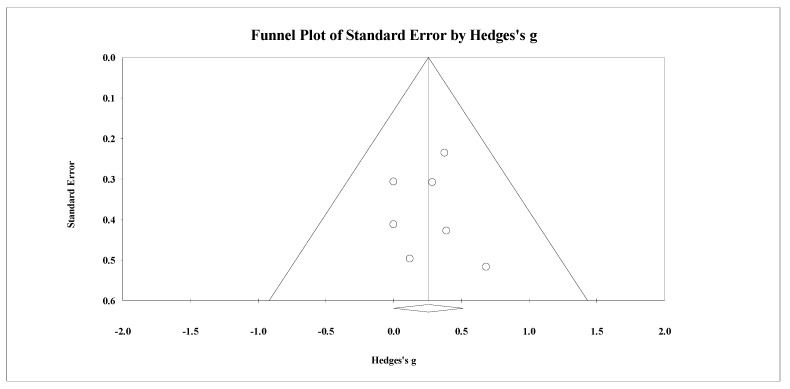
Funnel plot for changes in lower-body bone mineral density in untrained adults after a training program based on soccer small-sided games compared to a passive control condition. White circles: observed studies.

**Figure 10 healthcare-09-00457-f010:**
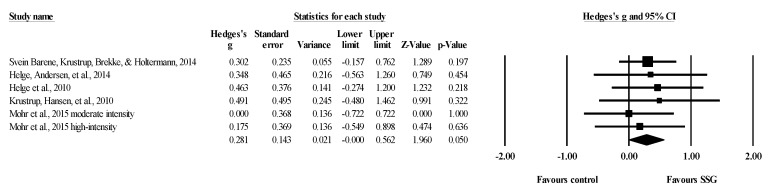
Forest plot of changes in lower-body bone mineral density in untrained adults after a training program based on soccer small-sided games (SSG) compared to active controls. Values shown are effect sizes (Hedges’s g) with 95% confidence intervals (CI). The size of the plotted squares reflects the statistical weight of each study. The black diamond reflects the overall result.

**Figure 11 healthcare-09-00457-f011:**
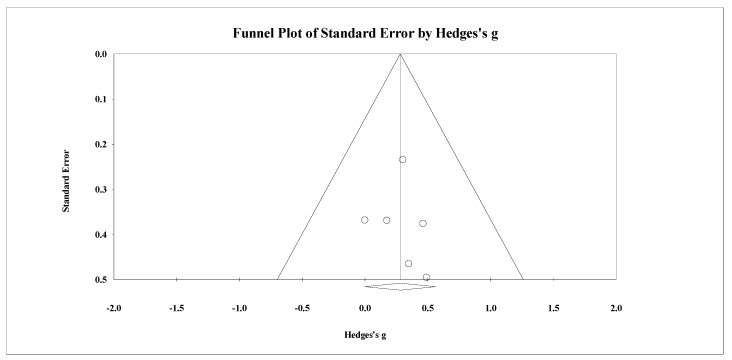
Funnel plot for changes in lower-body bone mineral density in untrained adults after a training program based on soccer small-sided games compared to an active control condition. White circles: observed studies.

**Figure 12 healthcare-09-00457-f012:**
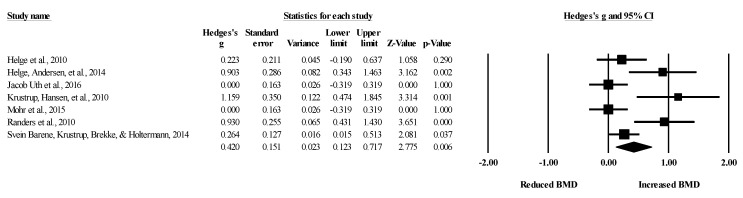
Forest plot of within-group changes in lower body bone mineral density (BMD) in untrained adults after a training program based on soccer small-sided games. Values shown are effect sizes (Hedges’s g) with 95% confidence intervals (CI). The size of the plotted squares reflects the statistical weight of each study. The black diamond reflects the overall result.

**Figure 13 healthcare-09-00457-f013:**
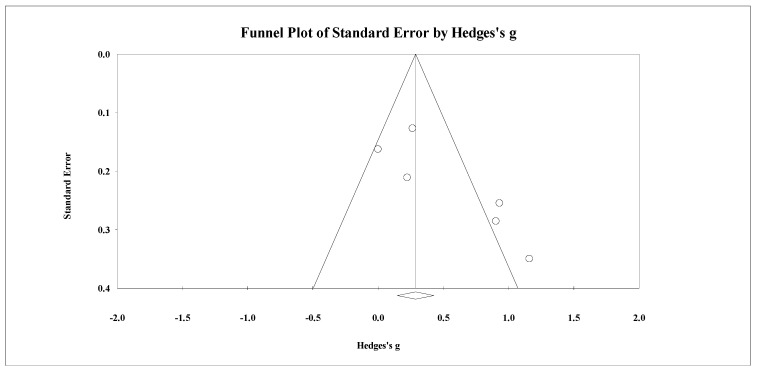
Funnel plot for changes in lower-body bone mineral density in untrained adults after a training program based on soccer small-sided games. White circles: observed studies.

**Table 1 healthcare-09-00457-t001:** Inclusion and exclusion criteria.

Item	Inclusion Criteria	Exclusion Criteria
Population	Untrained adults (>18 years old) from any sex, with or without a noncommunicable disease. Adults were not exposed to specific pharmacological or diet-oriented plans	Trained adults, athletes, youth (above 18 years old); participants were not exposed to specific pharmacological or diet-oriented plans
Intervention	SSG-based programs restricted to a minimum of 4 weeks (duration) and no restricted to frequency (number of sessions per week)	Other types of exercises; other types of SSGs; combined interventions (SSG and other types of exercise or intervention); or regular full-sized game (11 vs. 11); interventions with less than 4 weeks
Comparator	Passive or active control groups	Passive control with evidence of participation in structured exercise
Outcome	Pre-post intervention values (mean and standard deviation) of bone mineral density (BMD) measured in any body part	Other outcomes no, including bone mineral density (e.g., bone turnover markers); no information pre-post intervention (e.g., follow-up excluded); pre-post data the same in more than one article
Study Design	Randomized controlled trials	Nonrandomized studies
Additional criteria	Only original and full-text studies written in English	Written in another language than English; other article types than original (e.g., reviews, letters to editors, trial registrations, proposals for protocols, editorials, book chapters and conference abstracts)

**Table 2 healthcare-09-00457-t002:** Characteristics of the included studies and outcomes extracted.

Study	N	Mean Age (y)	Sex	Population	Type of Control Group	Test Used for Assessing the BMD	Body Parts Analyzed	Body Part Extracted for Meta-Analysis as Main Outcome
Barene et al. [43]	SSGs group: 37 Zumba group: 35 Passive control: 35	SSGs group: 44.1 ± 8.7 Zumba group: 45.9 ± 9.6 Passive control: 47.4 ± 9.5	Women ^a^	Health female hospital employees	One active control group (Zumba) One passive control group	DXA	Total body (g/cm^2^) Lumbar spine (g/cm^2^) Lower limb (g/cm^2^)	Total body (g/cm^2^) Lumbar spine (g/cm^2^) Lower limb (g/cm^2^)
Helge et al. [44]	SSGs group: 9 Resistance training: 8 Passive control: 6	SSGs group: 68.0 ± 4.0 Resistance training: 69.1 ± 3.1 Passive control: 67.4 ± 2.7	Men	Healthy elderly	One active control group (resistance training) One passive control group	DXA	Femoral neck (g/cm^2^) Femoral shaft (g/cm^2^) Proximal femur (g/cm^2^) Whole body (g/cm^2^)	Whole body (g/cm^2^) Femoral shaft (right) (g/cm^2^)
Helge et al. [45]	SSGs group: 12 Running group: 16 Passive control: 9	All: 36.3 ± 8.2	Women	Premenopausal women	One active control group (running training) One passive control group	pQCT	Total distal tibia (mg/cm^3^) Trabecular distal tibia (mg/cm^3^) Cortical + subcortical distal tibia (mg/cm^3^)	Total distal tibia right (mg/cm^3^)
Krustrup et al. [46]	SSGs group: 7 Running group: 8 Passive control: 7	SSGs group: 40 ± 3 Running group: 40 ± 2 Passive control: 38 ± 4	Women	Premenopausal women	One active control group (running training) One passive control group	DXA	Whole body (g/cm^2^) Legs (g/cm^2^)	Whole body (g/cm^2^) Legs (g/cm^2^)
Krustrup et al. [47]	SSGs group: 12 Running group: 9 Passive control: 10	SSGs group: 30 ± 2 Running group: 31 ± 2 Passive control: 30 ± 2	Men	Healthy	One active control group (running training) One passive control group	DXA	Whole body (g/cm^2^)	Whole body (g/cm^2^)
Mohr et al. [48]	SSGs group: 21 Swimming moderate: 21 Swimming high: 21 Passive control: 20	SSGs group: 45 ± 14 Swimming moderate: 46 ± 9 Swimming high: 44 ± 9 Passive control: 45 ± 9	Women	Premenopausal women	Two active control group (swimming at moderate and high-intensity) One passive control group	DXA	Total body (g/cm^2^) Leg right (g/cm^2^) Pelvis (g/cm^2^) Arm (g/cm^2^) Head (g/cm^2^)	Total body (g/cm^2^) Leg right (g/cm^2^) Pelvis (g/cm^2^)
Randers et al. [49]	SSGs group: 12 Passive control: 10	All: 20–43	Men	Healthy	One passive control group	DXA	Total body (g/cm^2^) Leg (g/cm^2^)	Total body (g/cm^2^) Leg (g/cm^2^)
Skoradal et al. [50]	SSGs group: 32 (14 women) Passive control: 23 (11 women)	All: 61 ± 9	Men and women	Prediabetes	One passive control group	DXA	Whole body (g/cm^2^) Head (g/cm^2^) Chest (g/cm^2^) Midriff (g/cm^2^) Pelvis (g/cm^2^) Arms (g/cm^2^)	Whole body (g/cm^2^) Midriff (g/cm^2^) Pelvis (g/cm^2^)
Uth et al. [51]	SSGs group: 21 Passive control: 20	SSGs group: 67.1 ± 7.1 Passive control: 66.5 ± 4.9	Men	Prostate cancer	One passive control group	DXA	Total body (g/cm^2^) Leg (g/cm^2^)	Total body (g/cm^2^) Leg (g/cm^2^)

N: number of participants in the study; M: men; W: women; h: hour; CG: control group; BMD: bone mineral density; a: only women were included in data analysis; DXA: dual-energy X-ray absorptiometry; SSG: small-sided games; pQCT: peripheral quantitative computed tomography.

**Table 3 healthcare-09-00457-t003:** Characteristics of small-sided games (SSG)-based programs in the included studies.

Study	Duration (W)	d/w	Session/Person Per Week (n)	Total Sessions	SSG Formats	SSG Pitch Dimension (Length × Width)	SSG Area Per Player (m^2^)	Sets	Reps	Recovery between Sets (min)	Recovery between Sets (Intensity)	Total Work Duration (min)	Work Duration Per Set (min)	Work Intensity (HRmax)
Barene et al. [43]	40 W	2–3	55.4 ± 8.8	-	3 vs.3 to 4 vs. 4	10 × 20 m	25 to 33	2	ND	5	-	50	25	78.6 ± 3.2%
Helge et al. [44]	52 W	ND	1.7 ± 0.3	ND	ND	ND	ND	3–4	ND	2	-	45	15	82%
Helge et al. [45]	14 W	2	ND	28.8 ± 5.0	5 vs. 5, 7 vs. 7 and 9 vs.9	30 to 40 wide to 45 to 60 m long	ND	4	ND	ND	ND	48	12	83 ± 0%
Krustrup et al. [46]	16 W	2	1.9 ± 0.1 to 1.7 ± 0.2	ND	4 vs. 4 to 5 vs. 5	20 to 30 wide × 30 to 40 m long	ND	ND	ND	ND	ND	60	ND	82 ± 2%
Krustrup et al. [47]	12 W	2–3	2.3	27.6	5 vs. 5 to 7 vs. 7	40 × 60 m	171 to 240	3–4	ND	ND	ND	60	ND	82 ± 2%
Mohr et al. [48]	15 W	3	3.0 ± 0.5	45 ± 5	4 vs. 4 to 10 vs. 10	ND	ND	ND	ND	ND	ND	60	ND	ND
Randers et al. [49]	64 W	ND	1.3–2.4	28.5 + 66.7	4 vs. 4 to 5 vs. 5	25 to 40 wide × 30 to 50 m long	ND	ND	ND	ND	ND	60	ND	81 ± 3 to 82 ± 2
Skoradal et al. [50]	16 W	2	2.0 ± 0.1	32 ± 2	4 vs. 4 to 6 vs. 6	ND	ND	2	ND	2–3	ND	30–60	15–30	79 ± 1
Uth et al. [51]	12 W	2–3	ND	20.6 ± 8.0	3 vs. 3 to 7 vs. 7	25 × 50 m for 6vs.6	100	2–3	ND	ND	ND	30–45	15	ND

SSGs: small-sided games; W: weeks; d/w: days per week; NR: not reported; m: meters; s: seconds; min: minutes; HRmax: maximal heart rate; ND: not described.

**Table 4 healthcare-09-00457-t004:** Characteristics of control groups.

Study	Active Control	Passive Control
Barene et al. [43]	Continuous dance movement using Latin music with varying intensity.	Only measurements were made; no intervention
Helge et al. [44]	Resistance training: 5 min low-intensity warm-up, followed by leg press, seated leg extension, hamstring curl, pull-down, and lateral dumbbell raises. Sets were interspaced by 1.5 min rest, and at the end of the session, 5 min of core training was made. Exercise progressed from 3 × 16–20RM (week 0–4), 3 × 12RM (week 5–8), 3 × 10RM (week 9–12) and 4 × 8RM (week13–52).	Inactive; no details
Helge et al. [45]	Running group: 5 min of low-intensity warm-up, followed by 4 × 12 min of continuous running and moderate intensity. After the 6 weeks, all runners were able to run for 55 min continuously.	Inactive; no details
Krustrup et al. [46]	One hour of running two times a week. Running speed was adjusted to fit 81% HRmax during the first 4 weeks and 82% in the last 12 months.	Continued daily live activities
Krustrup et al. [47]	The participants completed 3 to 4 sets within one hour of running, with an average intensity of 82% HRmax.	Continued daily live activities
Mohr et al. [48]	Moderate intensity swim: one hour per session, continuous front crawl swimming High-intensity swim: 15–25 min per session (3–5 min of effective swimming) consisting of 6–10 sets of 30 s bouts of all-out front crawl swimming with 2 min of passive recovery.	No training or lifestyle changes during the same period
Randers et al. [49]	-	Instructed to remain physically inactive
Skoradal et al. [50]	-	No details
Uth et al. [51]	-	Encouraged to maintain their normal level of physical activity

RM: repetition maximum; HRmax: maximum heart rate.

**Table 5 healthcare-09-00457-t005:** Physiotherapy evidence database (PEDro) scale ratings.

	No. 1 *	No. 2	No. 3	No. 4	No. 5	No. 6	No. 7	No. 8	No. 9	No. 10	No. 11	Total **
Barene et al. [43]	+	+	+	+	-	-	-	+	+	+	+	7
Helge et al. [44]	+	+	-	+	-	-	-	+	+	+	+	6
Helge et al. [45]	+	+	-	+	-	-	-	-	+	+	+	5
Krustrup et al. [46]	+	+	-	+	-	-	-	-	+	+	+	5
Krustrup et al. [47]	+	+	-	+	-	-	-	+	+	+	+	6
Mohr et al. [48]	+	+	-	+	-	-	-	+	+	+	+	6
Randers et al. [49]	+	+	-	+	-	-	-	-	+	+	+	5
Skoradal et al. [50]	+	+	-	+	-	-	-	+	+	+	+	6
Uth et al. [51]	+	+	-	+	-	-	-	+	+	+	+	6

*: PEDro scale items number; **: the total number of points from a possible maximal of 10; No. 1: eligibility criteria were specified (not included in the score); No. 2: subjects were randomly allocated to groups; No. 3: allocation was concealed; No. 4: the groups were similar at baseline regarding the most important prognostic indicators; No. 5: there was blinding of all subjects; No. 6: there was blinding of all therapists, who administered the therapy; No. 7: there was blinding of all assessors, who measured at least one key outcome; No. 8: measures of at least one key outcome were obtained from more than 85% of the subjects initially allocated to groups; No. 9: all subjects for whom outcome measures were available received the treatment or control condition as allocated or, where this was not the case, data for at least one key outcome was analyzed by “intention to treat”; No. 10: the results of between-group statistical comparisons are reported for at least one key outcome; and No. 11: the study provides both point measures and measures of variability for at least one key outcome.

**Table 6 healthcare-09-00457-t006:** Summary of the included studies and results of total body bone mineral density (BMD) before and after the intervention.

Study	Group	Sex	N	Before Mean ± SD	After Mean ± SD	After–Before (∆%)
Barene et al. [43]	SSG	W	37	1.12 ± 0.09	1.12 ± 0.10	0.0
Helge et al. [44]	SSG	M	9	1.17 ± 0.04	1.21 ± 0.04	3.4
Krustrup et al. [46]	SSG	W	7	1.22 ± 0.03	1.23 ± 0.03	0.8
Krustrup et al. [47]	SSG	M	12	1.24 ± 0.03	1.25 ± 0.02	0.8
Mohr et al. [48]	SSG	W	21	1.00 ± 0.08	0.99 ± 0.08	−1.0
Randers et al. [49]	SSG	M	12	1.30 ± 0.02	1.32 ± 0.02	1.5
Skoradal et al. [50]	SSG	M&W	32	1.01 ± 0.02	1.01 ± 0.02	0.0
Uth et al. [51]	SSG	M	21	1.17 ± 0.11	1.17 ± 0.11	0.0
Barene et al. [43]	AC	W	35	1.11 ± 0.08	1.11 ± 0.08	0.0
Helge et al. [44]	AC	M	8	1.23 ± 0.02	1.23 ± 0.02	0.0
Krustrup et al. [46]	AC	W	8	1.16 ± 0.03	1.16 ± 0.02	0.0
Krustrup et al. [47]	AC	M	9	1.33 ± 0.03	1.33 ± 0.03	0.0
Mohr et al. [48]	AC1	W	21	1.00 ± 0.12	1.00 ± 0.11	0.0
Mohr et al. [48]	AC2	W	21	1.00 ± 0.08	0.98 ± 0.08	−2.0
Randers et al. [49]	PC	M	10	1.26 ± 0.03	1.28 ± 0.03	1.6
Skoradal et al. [50]	PC	M&W	23	1.02 ± 0.04	1.03 ± 0.03	1.0
Uth et al. [51]	PC	M	20	1.21 ± 0.14	1.20 ± 0.13	−0.8
Barene et al. [43]	PC	W	35	1.11 ± 0.10	1.10 ± 0.10	−0.9
Helge et al. [44]	PC	M	6	1.27 ± 0.03	1.27 ± 0.03	0.0
Krustrup et al. [46]	PC	W	7	1.19 ± 0.04	1.21 ± 0.04	1.7
Krustrup et al. [47]	PC	M	10	1.28 ± 0.03	1.27 ± 0.03	−0.8
Mohr et al. [48]	PC	W	20	1.01 ± 0.09	0.99 ± 0.08	−2.0

SSG: small-sided game based-program; CG: control group; W: women; M: men; AC: active control; PC: passive control; AC1: swimming moderate intensity; AC2: swimming high-intensity; ∆%: percent changes representing mean differences (after–before).

**Table 7 healthcare-09-00457-t007:** Summary of the included studies and results of spine BMD before and after the intervention.

Study	Group	Sex	N	Before Mean ± SD	After Mean ± SD	After–before ∆(%)
Barene et al. [43]	SSG	W	37	1.12 ± 0.19	1.11 ± 0.17	−0.9%
Skoradal et al. [50]	SSG	M and W	32	1.38 ± 0.04	1.32 ± 0.04	−4.3
Barene et al. [43]	AC	W	35	1.07 ± 0.13	1.07 ± 0.16	0.0%
Barene et al. [43]	PC	W	35	1.10 ± 0.19	1.09 ± 0.18	−0.9%
Skoradal et al. [50]	PC	M&W	23	1.36 ± 0.07	1.33 ± 0.07	−2.2

SSG: small-sided game based-program; CG: control group; W: women; M: men; AC: active control; PC: passive control; AC1: swimming moderate intensity; AC2: swimming high-intensity; ∆%: percent changes representing mean differences (after–before).

**Table 8 healthcare-09-00457-t008:** Summary of the included studies and results of pelvis BMD before and after the intervention.

Study	Group	Sex	N	Before Mean ± SD	After Mean ± SD	After–before (∆%)
Mohr et al. [48]	SSG	W	21	1.06 ± 0.11	1.04 ± 0.11	−1.9
Skoradal et al. [50]	SSG	M&W	32	1.12 ± 0.03	1.11 ± 0.03	−0.9
Mohr et al. [48]	AC1	W	21	1.12 ± 0.21	1.08 ± 0.20	−3.6
Mohr et al. [48]	AC2	W	21	1.04 ± 0.12	1.03 ± 0.13	−1.0
Mohr et al. [48]	PC	W	20	1.09 ± 0.11	1.07 ± 0.13	−1.8
Skoradal et al. [50]	PC	M&W	23	1.14 ± 0.05	1.13 ± 0.05	−0.9

SSG: small-sided game based-program; CG: control group; W: women; M: men; AC: active control; PC: passive control; AC1: swimming moderate intensity; AC2: swimming high-intensity; ∆%: percent changes representing mean differences (after–before).

**Table 9 healthcare-09-00457-t009:** Summary of the included studies and results of lower limb BMD before and after the intervention.

Study	Group	Sex	N	Before Mean ± SD	After Mean ± SD	After–Before (∆%)
Barene et al. [43]	SSG	W	37	2.24 ± 0.18	2.29 ± 0.19	2.2
Helge et al. [44]	SSG	M	9	1.12 ± 0.04	1.16 ± 0.04	3.6
Helge et al. [45]	SSG	W	12	301.4 ± 29.6	307.6 ± 28.7	2.1
Krustrup et al. [46]	SSG	W	7	1.32 ± 0.03	1.36 ± 0.03	3.0
Mohr et al. [48]	SSG	W	21	1.04 ± 0.08	1.04 ± 0.07	0.0
Randers et al. [49]	SSG	M	12	1.53 ± 0.03	1.56 ± 0.03	2.0
Uth et al. [51]	SSG	M	21	1.28 ± 0.13	1.28 ± 0.13	0.0
Barene et al. [43]	AC	W	35	2.26 ± 0.24	2.26 ± 0.23	0.0
Helge et al. [44]	AC	M	8	290.3 ± 19.5	293.6 ± 21.1	1.1
Helge et al. [45]	AC	W	16	1.24 ± 0.06	1.23 ± 0.06	−0.8
Krustrup et al. [46]	AC	W	8	1.23 ± 0.02	1.26 ± 0.01	2.4
Mohr et al. [48]	AC1	W	21	1.05 ± 0.09	1.05 ± 0.09	0.0
Mohr et al. [48]	AC2	W	21	1.04 ± 0.07	1.03 ± 0.07	−1.0
Barene et al. [43]	PC	W	35	2.29 ± 0.21	2.28 ± 0.23	−0.4
Helge et al. [44]	PC	M	6	279.1 ± 34.1	276.5 ± 32.3	−0.9
Helge et al. [45]	PC	W	9	1.24 ± 0.06	1.25 ± 0.06	0.8
Krustrup et al. [46]	PC	W	7	1.31 ± 0.04	1.33 ± 0.04	1.5
Mohr et al. [48]	PC	W	20	1.05 ± 0.10	1.03 ± 0.10	−1.9
Randers et al. [49]	PC	M	10	1.41 ± 0.03	1.44 ± 0.04	2.1
Uth et al. [51]	PC	M	20	1.31 ± 0.16	1.31 ± 0.16	0.0

SSG: small-sided game based-program; CG: control group; ∆%: percent changes representing mean differences (after–before).

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
