# Peer review of "Effects of Recreational Small-Sided Soccer Games on Bone Mineral Density in Untrained Adults: A Systematic Review and Meta-Analysis"

_healthcare, 2021, doi:10.3390/healthcare9040457_

Round 1

Reviewer 1 Report

The authors performed systematic review with meta-analysis to assess the effects of small-sided soccer game (SSG)-based training programs on bone mineral density (BMD). They found that SSG significantly increases BMD in lower limbs, although SSG does not significantly increase total BMD. 

They summarized and analyzed previous studies in depth. The manuscript is properly organized and well written.

I have some suggestions that could improve the manuscript.

  1. The definition of SSG should be described more clearly, although there seems to be some variation in SSG as described in Table 3.
  2. If there are available comparative data of BMD between amateur soccer players, who play soccer regularly, and age- and sex-matched controls, the data should be included. The difference of the values of BMD would reflect the possible maximum effect of soccer-based training on BMD.    
  3. Please delete % in the values of after-before to be consistent with other tables.

Author Response

REVIEWER 1

The authors performed systematic review with meta-analysis to assess the effects of small-sided soccer game (SSG)-based training programs on bone mineral density (BMD). They found that SSG significantly increases BMD in lower limbs, although SSG does not significantly increase total BMD. 

They summarized and analyzed previous studies in depth. The manuscript is properly organized and well written.

AUTHORS: DEAR REVIEWER, THANK YOU SO MUCH FOR YOUR COMMENTS AND SUGGESTIONS. WE HAVE HIGHLIGHTED THE CHANGES RELATED TO YOUR REVISIONS IN GREEN.

I have some suggestions that could improve the manuscript.

  1. The definition of SSG should be described more clearly, although there seems to be some variation in SSG as described in Table 3.

AUTHORS: DEAR REVIEWER, THANK YOU. WE HAVE DETAILED THE MEANING OR SSG IN THE INTRODUCTION (5TH PARAGRAPH).

  1. If there are available comparative data of BMD between amateur soccer players, who play soccer regularly, and age- and sex-matched controls, the data should be included. The difference of the values of BMD would reflect the possible maximum effect of soccer-based training on BMD.

AUTHORS: DEAR REVIEWER, THANK YOU. HOWEVER, OUR TARGET WAS TO FOCUS ONLY IN UNTRAINED POPULATIONS, AIMING TO IDENTIFY HOW RECREATIONAL SOCCER CAN BE INTERESTING AS ALTERNATIVE APPROACH TO OTHER EXERCISE MODE.    

  1. Please delete % in the values of after-before to be consistent with other tables.

AUTHORS: DEAR REVIEWER, THANK YOU. WE NEED TO KEEP THE “%” SINCE THE COLUMN REPRESENT THE PERCENTUAL VARIATION BETWEEN POST-PRE, WHILE THE OTHER COLUMNS ARE THE MEAN AND STANDARD DEVIATION. WE HAVE ADDED THIS INFORMATION IN THE CAPTION.

Reviewer 2 Report

General comments:

This is an interesting review topic that has great applicability to 'real world' practice. It is clear that considerable efforts have been invested in this project. But I think there are some improvements that could be made to enhance the linkages of certain points and also to highlight the novelty factors further. The introduction requires some adjustment to inform on the impact of small-sided soccer games on bone mineral density in untrained adults and the rationale for the outcome measures selected. The below comments are intended to explain this view and are hoped to be of benefit to the authors.

Specific comments

Title: It may be worth adjusting the title slightly by removing "randomized controlled trials" as this information has already been given in the abstract and the method section.

Abstract:

L22-23: The full names of SSG and BMD must be given as they appear for the first time.

L25: Please give the average age of the population as the mean±SD

Introduction:

L46-47: “This mechanism consists of a conversion of a mechanical force into a cellular response” this sentence needs to be expanded.

L47-49: The authors reported interesting findings from literature on the relationship between the mechanical load and bone remodeling, however the bone's structural adaptations to mechanical load should be more developed, authors can recall Wolff's law… It is necessary to include this part to explain the backdrop of what is already known/not known.

L55: Add a reference please

L60-62: "it seems that specific exercises (e.g., swimming or cycling) are not meaningfully effective in menopausal women or even in children and adolescents", But authors should recall also that physical activities with a high mechanical impact allow an increase in BMD and stimulate the mineralization and synthesis of the bone matrix due to strong mechanical stresses exerted on the bone tissue.

-Linking sentences need to be improved.

 Methods:

-The information sources and the description of literature search (which database was used, publication date interval (from which-to-which year), the used keys words ... and whether the authors do a manual search to obtain further studies not identified electronically) should be presented as the first part in the method section before the eligibility criteria. should be presented as the first part of the method section before the eligibility criteria. To comply with the PRISMA guidelines, it is necessary to begin with the literature search.

-L118: The last literature search was conducted on January 16 (2021). May be an updated search is needed.

Summary measures, synthesis of results, and publication bias:

L162-163: Authors have used the Kappa correlation test to analyze the agreement level for the included studies. Add please these results as a supplementary data

L179-182: In the case of medium or high heterogeneity, did the authors do further investigations through a subgroup analysis of moderator variables?

Results:

-Authors have selected the sex, the age and the exercise intensity as moderators but they did not report the differences between “men and women” or “old and young” or “low and heavy work intensity” in experimental groups, please add these results in the manuscript or as supplementary data.

-Authors presented the difference between “experimental group” and “passive group” or “active group” But they did not describe sufficiently the difference between passive group and active group (did the active group undertake a low or moderate physical program?) please add more details

L311-322: The results of sections (3.5) and (3.6) have not been sufficiently described.

Discussion:                                                                                                                                                              

-Large sections throughout are written in narrative form and do not address the topics typically expected of review of the submitted type. For example, discussion should focus on the outcomes of the current results, overall completeness and applicability of evidence, quality of the evidence, potential biases in the review process, and agreements and disagreements with other studies or reviews. The discussion requires considerable redrafting to align with the review type identified.

L396: “it does not provide enough stimulation to improve total body BMD in untrained adults” This study includes only 10 studies so the authors cannot give any evidence or confirmation, they should add "may" or "could" to this sentence.

L397-400: “These results could be influenced by………………….. duration of 16 months” These results are influenced by a single study finding, authors should add this as a limitation as we cannot draw robust conclusions based on a single study.

L495-515: The mechanistic discussion of this part is weak if almost absent. Please consider including possible mechanisms improving the BMD duo SSGs-based interventions. The authors recite the study findings in the discussion part, but they do not really discuss these results and they do not compare their results with what is observed in previous studies.

Author Response

REVIEWER 2

General comments:

This is an interesting review topic that has great applicability to 'real world' practice. It is clear that considerable efforts have been invested in this project. But I think there are some improvements that could be made to enhance the linkages of certain points and also to highlight the novelty factors further. The introduction requires some adjustment to inform on the impact of small-sided soccer games on bone mineral density in untrained adults and the rationale for the outcome measures selected. The below comments are intended to explain this view and are hoped to be of benefit to the authors.

AUTHORS: DEAR REVIEWER, THANK YOU SO MUCH FOR YOUR IMPORTANT COMMENTS AND SUGGESTIONS. WE HAVE MADE CHANGES FOLLOWING YOUR COMMENTS. ALL THE CHANGES RELATED TO YOUR REVISIONS CAN BE OSERVED IN BLUE.

Specific comments

Title: It may be worth adjusting the title slightly by removing "randomized controlled trials" as this information has already been given in the abstract and the method section.

AUTHORS: DEAR REVIEWER, THANK YOU. WE HAVE REMOVED.

Abstract:

L22-23: The full names of SSG and BMD must be given as they appear for the first time.

AUTHORS: DEAR REVIEWER, THANK YOU. WE HAVE CHANGED ACCORDINGLY.

L25: Please give the average age of the population as the mean±SD

AUTHORS: DEAR REVIEWER, THANK YOU. WE HAVE ADDED THE RANGE OF INCLUDED POPULATION.

Introduction:

L46-47: “This mechanism consists of a conversion of a mechanical force into a cellular response” this sentence needs to be expanded.

AUTHORS: DEAR REVIEWER, THANK YOU. WE HAVE ADDED MORE INFORMATION IN THE PARAGRAPH.

L47-49: The authors reported interesting findings from literature on the relationship between the mechanical load and bone remodeling, however the bone's structural adaptations to mechanical load should be more developed, authors can recall Wolff's law… It is necessary to include this part to explain the backdrop of what is already known/not known.

AUTHORS: DEAR REVIEWER, THANK YOU. WE HAVE WORKED ON THE PARAGRAPH.

L55: Add a reference please

AUTHORS: DEAR REVIEWER, THANK YOU. WE HAVE ADDED.

L60-62: "it seems that specific exercises (e.g., swimming or cycling) are not meaningfully effective in menopausal women or even in children and adolescents", But authors should recall also that physical activities with a high mechanical impact allow an increase in BMD and stimulate the mineralization and synthesis of the bone matrix due to strong mechanical stresses exerted on the bone tissue.

AUTHORS: DEAR REVIEWER, THANK YOU. WE HAVE FOLLOWED YOUR SUGGESTION.

-Linking sentences need to be improved.

AUTHORS: DEAR REVIEWER, THANK YOU. WE HAVE WORKED ON THE LINKS BETWEEN PARAGRAPHS.

 Methods:

-The information sources and the description of literature search (which database was used, publication date interval (from which-to-which year), the used keys words ... and whether the authors do a manual search to obtain further studies not identified electronically) should be presented as the first part in the method section before the eligibility criteria. should be presented as the first part of the method section before the eligibility criteria. To comply with the PRISMA guidelines, it is necessary to begin with the literature search.

AUTHORS: DEAR REVIEWER, THANK YOU. HOWEVER, IN THE PRISMA CHECKLIST 2009 THE ELIGIBILITY CRITERIA (ITEM 6) FOLLOWS IMMEDIATELY OF PROTOCOL REGISTRATION (ITEM 5) AND BEFORE THE INFORMATION SOURCES (ITEM 7) AND SEARCH (ITEM 8). WE HAVE TRIED TO FOLLOW THE SEQUENCE.

-L118: The last literature search was conducted on January 16 (2021). May be an updated search is needed.

AUTHORS: DEAR REVIEWER, IN A BRIEF SEARCH AFTER THE PERIOD, NO ARTICLE FOR INCLUSION WAS FOUND. THUS, WE WILL KEEP THE INITIAL SEARCH, SINCE THE REVIEW HAS TWO MONTHS (NOT TOO MUCH).

Summary measures, synthesis of results, and publication bias:

L162-163: Authors have used the Kappa correlation test to analyze the agreement level for the included studies. Add please these results as a supplementary data

AUTHORS: DEAR REVIEWER, THANK YOU. WE HAVE ADDED THE EXACT VALUE OBTAINED FROM CROSSTABULATION IN OVERALL RESULTS (k=0.859)                                 

L179-182: In the case of medium or high heterogeneity, did the authors do further investigations through a subgroup analysis of moderator variables?

AUTHORS: DEAR REVIEWER, THANK YOU. DUE TO THE FACT THE ANALYSIS WAS MADE BETWEEN EXERCISE VS. CONTROL VS. ACTIVE GROUP, THE NUMBER FOR MODERATOR ANALYSIS WAS NOT ENOUGH.

Results:

-Authors have selected the sex, the age and the exercise intensity as moderators but they did not report the differences between “men and women” or “old and young” or “low and heavy work intensity” in experimental groups, please add these results in the manuscript or as supplementary data.

AUTHORS: DEAR REVIEWER, THANK YOU. THE SUB-GROUP ANALYSIS WAS NOT POSSIBLE TO COMPUTE DUE TO THE LACK OF ACCEPTABLE NUMBER OF GROUPS PER MODERATOR (3 PER EACH MODERATOR AND GROUP). THUS WAS NOT POSSIBLE TO CONSIDERE MODERTORS IN THE ANALYSIS.

-Authors presented the difference between “experimental group” and “passive group” or “active group” But they did not describe sufficiently the difference between passive group and active group (did the active group undertake a low or moderate physical program?) please add more details

AUTHORS: DEAR REVIEWER, THANK YOU. WE HAVE ADDED A NEW TABLE (N.º4) WITH DETAILS ABOUT CONTROL GROUPS.

L311-322: The results of sections (3.5) and (3.6) have not been sufficiently described.

AUTHORS: DEAR REVIEWER, THANK YOU. WE HAVE ADDED A DESCRIPTION OF RESULTS.

Discussion:                                                                                                                                                              

-Large sections throughout are written in narrative form and do not address the topics typically expected of review of the submitted type. For example, discussion should focus on the outcomes of the current results, overall completeness and applicability of evidence, quality of the evidence, potential biases in the review process, and agreements and disagreements with other studies or reviews. The discussion requires considerable redrafting to align with the review type identified.

AUTHORS: DEAR REVIEWER, THANK YOU. WE HAVE MODIFIED THE DISCUSSION SECTION ACCORDING TO THE REVIEWER COMMENTS.

L396: “it does not provide enough stimulation to improve total body BMD in untrained adults” This study includes only 10 studies so the authors cannot give any evidence or confirmation, they should add "may" or "could" to this sentence.

AUTHORS: DEAR REVIEWER, THANK YOU. THIS SENTENCE HAS BEEN MODIFIED.

L397-400: “These results could be influenced by………………….. duration of 16 months” These results are influenced by a single study finding, authors should add this as a limitation as we cannot draw robust conclusions based on a single study.

AUTHORS: DEAR REVIEWER, THANK YOU. THIS HAS BEEN ADDED AS A LIMITATION.

L495-515: The mechanistic discussion of this part is weak if almost absent. Please consider including possible mechanisms improving the BMD duo SSGs-based interventions. The authors recite the study findings in the discussion part, but they do not really discuss these results and they do not compare their results with what is observed in previous studies.

AUTHORS: DEAR REVIEWER, THANK YOU. WE UNDERSTAND THE REVIEWER CONCERN. HOWEVER, WE CONSIDER THAT THIS SUB-SECTION (4.4.    Study limitations, future research and clinical implications) MUST SHOW INFORMATION INSTEAD TO DISCUSS THE RESULTS, WHICH HAS BEEN PREVIOUSLY CARRY OUT.

Round 2

Reviewer 2 Report

The authors resubmitted their study focus on “Effects of recreational small-sided soccer games on bone mineral density in untrained adults: A systematic review and meta-analysis”. Overall, the authors have taken into consideration all the suggestions and have massively enhanced the quality of the manuscript, but the manuscript still needs improvements to be publish in Health Care. The below comments are intended to explain this view and are hoped to be of benefit to the authors.

Methods:

-The last literature search was conducted on January 16 (2021), an updated search is needed. The authors have mentioned that there is no article found for inclusion, even if there is no study eligible for inclusion, authors should add the number of excluded studies and their reasons for exclusion in the PRISMA flow diagram.

-I suggest that the authors add a Funnel plots to identify a possible publication bias for more visibility on the publications that tend to be distributed toward the positive effect

Discussion:

-The mechanistic discussion of the last part of the discussion is weak if almost absent, authors must include possible mechanisms improving the BMD duo SSGs-based interventions.

Limitations:

-This study includes only 10 studies, and some findings were based on a single study. so we cannot draw a robust conclusion, or an evidence based on these findings. 

-Authors did not provide any comparison between the moderator subgroup (sex, age, exercise intensity), Authors should include this as a limitation.

Author Response

REVIEWER 2

The authors resubmitted their study focus on “Effects of recreational small-sided soccer games on bone mineral density in untrained adults: A systematic review and meta-analysis”. Overall, the authors have taken into consideration all the suggestions and have massively enhanced the quality of the manuscript, but the manuscript still needs improvements to be publish in Health Care. The below comments are intended to explain this view and are hoped to be of benefit to the authors.

Methods:

-The last literature search was conducted on January 16 (2021), an updated search is needed. The authors have mentioned that there is no article found for inclusion, even if there is no study eligible for inclusion, authors should add the number of excluded studies and their reasons for exclusion in the PRISMA flow diagram.

AUTHORS: DEAR REVIWER, THANK YOU. WE HAVE UPDATED THE SEARCH AND THE FLOWCHART.

-I suggest that the authors add a Funnel plots to identify a possible publication bias for more visibility on the publications that tend to be distributed toward the positive effect

AUTHORS: DEAR REVIEWER, THANK YOU. WE HAVE ADDED THE FUNNEL PLOT.

Discussion:

-The mechanistic discussion of the last part of the discussion is weak if almost absent, authors must include possible mechanisms improving the BMD duo SSGs-based interventions.

AUTHORS: DEAR REVIEWER, THANK YOU. WE HAVE TRIED TO EXPAND IN SECTION 4.2

Limitations:

-This study includes only 10 studies, and some findings were based on a single study. so we cannot draw a robust conclusion, or an evidence based on these findings. 

AUTHORS: DEAR REVIEWER, THANK YOU. WE HAVE ADDED IN THE STUDY LIMITATIONS AND IN THE CONCLUSIONS SECTION.

-Authors did not provide any comparison between the moderator subgroup (sex, age, exercise intensity), Authors should include this as a limitation.

AUTHORS: DEAR REVIEWER, THANK YOU. WE HAVE ADDED IN THE STUDY LIMITATIONS.

Round 3

Reviewer 2 Report

Overall, the authors have taken into consideration all the suggestions and have massively enhanced the quality of the manuscript, but an extensive editing of English language and style is required  to be suitable for publication